# Fusion of tethered membranes can be driven by Sec18/NSF and Sec17/αSNAP without HOPS

Hongki Song, William T Wickner*

Department of Biochemistry and Cell Biology, Geisel School of Medicine at Dartmouth, Hanover, United States

**Abstract** Yeast vacuolar membrane fusion has been reconstituted with R, Qa, Qb, and Qc-family SNAREs, Sec17/αSNAP, Sec18/NSF, and the hexameric HOPS complex. HOPS tethers membranes and catalyzes SNARE assembly into RQaQbQc *trans*-complexes which zipper through their SNARE domains to promote fusion. Previously, we demonstrated that Sec17 and Sec18 can bypass the requirement of complete zippering for fusion (Song et al., 2021), but it has been unclear whether this activity of Sec17 and Sec18 is directly coupled to HOPS. HOPS can be replaced for fusion by a synthetic tether when the three Q-SNAREs are pre-assembled. We now report that fusion interme-diates with arrested SNARE zippering, formed with a synthetic tether but without HOPS, support Sec17/Sec18-triggered fusion. This zippering-bypass fusion is thus a direct result of Sec17 and Sec18 interactions: with each other, with the platform of partially zippered SNAREs, and with the apposed tethered membranes. As these fusion elements are shared among all exocytic and endocytic traffic, Sec17 and Sec18 may have a general role in directly promoting fusion.

## Editor's evaluation

This Research Advance answers an important question arising from the author's previous paper on in vitro vacuole fusion: whether the tether HOPS is needed for an unexpected direct role played by Sec17 and Sec18 in facilitating SNARE zippering and thus membrane fusion. An artificial tether substitutes for HOPS, so HOPS is not specifically needed. Therefore, this unexpected role for Sec17/18 could extend to other SNARE-mediated fusion reactions.

**\*For correspondence:**
Bill.Wickner@Dartmouth.edu

**Competing interest:** The authors declare that no competing interests exist.

## Introduction

Intracellular membrane fusion is catalyzed by families of fusion proteins which are conserved from yeast to humans and among the organelles (*Wickner and Rizo, 2017*). These include Rab-family GTPases, large tethering complexes which bind to Rabs (*Baker and Hughson, 2016*), membrane-anchored SNARE proteins (*Jahn and Scheller, 2006*), and SNARE chaperones of the SM, Sec17/αSNAP, and Sec18/NSF families. SNAREs have conserved SNARE domains of R, Qa, Qb, and Qc sub-families (*Fasshauer et al., 1998*) which assemble into tetrameric RQaQbQc coiled coils *trans*-complexes that bridge membranes before fusion. Each SNARE domain has heptad repeating apolar residues which are buried together in the center of the assembling four-helical coiled coils structure, driving SNARE assembly in an N- to C-direction. This directional assembly, termed 'zippering', can provide sufficient energy for fusion (*Sorensen et al., 2006*) and has been thought to be the sole driving force for fusion while other proteins simply catalyze or regulate SNARE assembly.

We study membrane fusion mechanisms with the vacuoles (lysosomes) of *Saccharomyces cere-visiae*. Yeast vacuole fusion requires proteins of each canonical conserved fusion-protein family, but

combines the Rab-effector tethering and the SM functions into the large hexameric HOPS (homotypic fusion and vacuole protein sorting) complex (*Wurmser et al., 2000*; *Seals et al., 2000*; *Stroupe et al., 2006*; *Hickey and Wickner, 2010*). HOPS binds to acidic lipids (*Orr et al., 2015*) and to its Rab Ypt7 on each fusion partner membrane (*Hickey and Wickner, 2010*), activating HOPS (*Torng et al., 2020*) to initiate N- to C-directional assembly among the R, Qa, Qb, and Qc SNAREs. HOPS has direct affinity for the R and Qa SNAREs through its SM subunit (*Baker et al., 2015*) and for the Qb SNARE (*Song et al., 2020*) and Qc SNARE (*Stroupe et al., 2006*) as well. These multiple affinities directly affect the $K_m$ of SNAREs for SNARE complex assembly (*Zick and Wickner, 2013*). HOPS can assemble intermediates formed from R and any two Q SNAREs which are triggered to undergo rapid fusion by the third Q SNARE (*Song et al., 2020*).

There has been extensive study, and evolving understanding, of how Sec17/αSNAP and Sec18/NSF promote membrane fusion. ATP hydrolysis by NSF/Sec18 was initially proposed to directly drive fusion (*Malhotra et al., 1988*), while SNAREs mediated the attachment of the NSF fusion engine to membranes (*Söllner et al., 1993b*) via αSNAP/Sec17 (*Whiteheart et al., 1992*). It was subsequently shown that Sec18/NSF and Sec17/αSNAP disassemble SNARE complexes (*Söllner et al., 1993a*), that Sec18/NSF and Sec17/αSNAP act prior to membrane docking (*Mayer et al., 1996*), that NSF/Sec18 and SNAP/Sec17 associate in a specific complex with the SNAREs and induce an conformational change in the SNAREs (*Hanson et al., 1995*; *Hanson et al., 1997*), and that SNARE complexes alone can mediate fusion (*Weber et al., 1998*). These findings led to the concept that all the energy for fusion derives from SNARE zippering (*Weber et al., 1998*; *Sorensen et al., 2006*). Though Sec17/αSNAP will block fusion from spontaneously assembled *trans*-SNARE complexes (*Mima et al., 2008*; *Stepien et al., 2019*), other fusion proteins such as vacuolar HOPS or the synaptic proteins Munc18-1, Munc13-1, complexin-1, and synaptotagmin confer resistance to Sec17/αSNAP and Sec18/NSF (*Mima et al., 2008*; *Prinslow et al., 2019*). Vacuolar SNAREs undergoing HOPS-dependent assembly are resistant to disassembly by Sec17/Sec18 (*Xu et al., 2010*), but Sec17 and Sec18 otherwise keep most SNAREs in a disassembled state; indeed most of the Qa and Qb SNAREs on isolated vacuoles are not together in complex (*Collins et al., 2005*).

Without Sec17/Sec18, HOPS-assembled *trans*-SNARE complexes require complete zippering to induce fusion, since fusion is arrested by deletion of the C-terminal 21 residues of the Qc SNARE domain (*Schwartz and Merz, 2009*), or the Qa or Qb SNARE domains (*Song et al., 2021*). This fusion arrest is bypassed by adding Sec17 (*Schwartz and Merz, 2009*; *Song et al., 2021*), driven by the apolar N-terminal loop of Sec17 (*Schwartz et al., 2017*). Fusion with only SNAREs is blocked by Sec17/Sec18 (*Mima et al., 2008*). HOPS not only prevents Sec17 inhibition of fusion, but acts synergistically with Sec17/Sec18 for fusion with wild-type SNAREs (*Mima et al., 2008*; *Schwartz et al., 2017*; *Song et al., 2017*). When fusion is blocked by zippering arrest due to deletions from the C-terminus of vacuolar Qc, the block can be bypassed by Sec17, both in vitro and in vivo (*Schwartz and Merz, 2009*; *Schwartz et al., 2017*). A similar block to fusion is also seen with C-terminal truncation of the Qa or Qb SNAREs, singly or in combinations (*Song et al., 2021*). Fusion is restored in each case by Sec17/Sec18 without requiring ATP hydrolysis. Even when both the Qb and Qc SNAREs were C-terminally truncated and the Qa SNARE was mutated to impart polarity to its canonical apolar SNARE domain residues, HOPS-dependent fusion was restored by Sec17/Sec18. Sec17 has direct affinity for HOPS (*Song et al., 2021*) through its Vps33 subunit (*Lobingier et al., 2014*); it has been unclear whether HOPS is needed for Sec17 and Sec18 to engage partially zippered SNAREs and mediate zippering-bypass fusion.

Sec17 and Sec18 enhance the rate of HOPS-dependent vacuole membrane fusion with all wild-type fusion components (*Mima et al., 2008*; *Zick et al., 2015*; *Song et al., 2017*; *Schwartz et al., 2017*) without requiring ATP hydrolysis (*Zick et al., 2015*; *Song et al., 2017*). A recent study (*Song et al., 2021*) has shown that Sec17, which promotes zippering (*Ma et al., 2016*), will also act through its N-terminal membrane-proximal apolar loop to drive HOPS-dependent fusion even when zippering would be unable to provide any fusion energy. Slow fusion can be driven either by zippering or by the apolar loops of Sec17 which had assembled on a platform of partially zippered *trans*-SNARE complex. Rapid HOPS-dependent fusion requires both SNARE zippering and Sec17 with its apolar loops, as analyzed in vitro and in vivo, where overexpression of Sec17 with an intact N-terminal apolar loop substantially restored the fusion defect caused by Qc SNARE domain truncation (*Schwartz et al., 2017*). Given the two complementary routes of complete SNARE zippering and Sec17 apolar loop

insertion, each capable of driving fusion, it is perhaps unsurprising that the precise starting conditions of fusion assays control when the contributions of zippering and of Sec17 are evident. The rate of fusion and its dependence on Sec17 are governed by the fluidity and headgroup composition of the membrane lipids and the initial state of assembly of the SNAREs (*Zick et al., 2015*). Sec17 is required when all the SNAREs are initially disassembled (*Zick et al., 2015*), as found on intact vacuoles (*Collins et al., 2005*) where only a tenth of the Qb SNARE is in complex with Qa (*Collins et al., 2005*), presumably due to continuous *cis*-SNARE disassembly in cells by Sec17/Sec18/ATP. With vacuolar mimic lipid composition and two of the Q SNAREs pre-assembled, fusion is stimulated several-fold by Sec17 and Sec18 (*Song et al., 2017*). When all three Q SNAREs are initially pre-assembled on one membrane, tethering is the only prerequisite for fusion and Sec17/Sec18 is not essential (*Baker et al., 2015*; *Song and Wickner, 2019*). Though little 3Q SNARE complex is found on intact vacuoles (*Collins et al., 2005*), pre-assembly of 3Q-SNARE complex during proteoliposome preparation allows synthetic tethers to support fusion as well as HOPS (*Song and Wickner, 2019*), which is useful for mechanistic studies. Sec17/Sec18 dependence can be restored to fusion reactions where the 3Q SNAREs had been pre-assembled by preventing zippering through the Qc3Δmutation. It has been unclear whether the actions of Sec17 and Sec18 to promote fusion are particular to HOPS-dependent fusion, and thus restricted to the fusion of vacuoles/lysosomes. Our current work, substituting a synthetic protein tether for HOPS, shows that Sec17/Sec18 can also drive HOPS-independent fusion.

We now exploit a synthetic tether to show that HOPS is not required for Sec17 and Sec18 to drive zippering-bypass fusion. The 4 SNAREs, Sec17/αSNAP, and Sec18/NSF, the fundamental components of the 20 s particle (*Zhao et al., 2015*), suffice to drive fusion. As a synthetic tether, we employ the dimeric protein glutathione-*S*-transferase (GST) fused to a PX domain. Dimeric GST-PX binds PI3P in each bilayer to tether membranes (*Song and Wickner, 2019*). When Q SNAREs are pre-assembled into a QaQbQc ternary complex on one fusion partner membrane, tethering by GST-PX supports fusion with membranes bearing the R SNARE without the need for SM function, Sec17 or Sec18 (*Song and Wickner, 2019*). This fusion relies on SNARE zippering, as it is blocked by deletion of the C-terminal region of the Qc SNARE domain, the Qc3Δ mutation. We find that GST-PX tethered membranes bearing R and QaQbQc3Δ SNAREs on fusion partners, unable to completely zipper and fuse because of the shortened Qc SNARE domain, are rescued from this arrested state and will fuse upon addition of Sec17 and Sec18. Thus, the Sec17 and Sec18 fusion functions do not rely on interactions with HOPS, instead acting through their interactions with each other, with tethered membranes, and with a partially zippered *trans*-SNARE binding platform.

## Results

Proteoliposomes were prepared with vacuolar lipids, with membrane-anchored Ypt7, and with either the R- or the 3Q SNAREs. The Qc SNARE was either wild type with its full-length SNARE domain or Qc3Δ which lacks the C-terminal four heptads of its SNARE domain. The Ypt7/R- and Ypt7/3Q-proteoliposomes bore lumenal fusion-reporter fluorescent proteins, either Cy5-labeled streptavidin or biotinylated phycoerythrin (*Figure 1A*). Proteoliposomes were purified by flotation to remove unincorporated proteins. When these proteoliposomes are mixed, their lumenal fluorescent proteins are separated by at least the thickness of two lipid bilayers, too far for measurable fluorescence resonance energy transfer (FRET). Upon fusion and the attendant content mixing, the binding of biotin to streptavidin brings the Cy5 and phycoerythrin fluorophores into intimate contact, yielding a strong FRET signal (*Zucchi and Zick, 2011*). Fusion incubations were performed with mixed Ypt7/R and Ypt7/3Q proteoliposomes and with external nonfluorescent streptavidin to block any signal from proteoliposome lysis. Each incubation had either HOPS or GST-PX to tether the membranes (*Figure 1A*). Also present from the start were either (a) buffer alone, (b) Sec17, (c) Sec18 and ATPγS, or (d) both Sec17 and Sec18/ATPγS. Fusion was monitored by FRET between the lumenal probes; the initial rate during the first 5 min is termed the α portion in *Figure 1B–E*. At 30 min, each reaction received a supplement of the components not added at time 0, that is, (a) Sec17, Sec18 and ATPγS, (b) Sec18 and ATPγS, (c) Sec17, or (d) buffer alone, so that all incubations had Sec17, Sec18, and ATPγS as the incubation continued in the β portion of the experiment (*Figure 1B* β-Eβ), from 30 to 32 min. Distinct information can be gleaned from the α and β intervals of the experiment, and these are considered in turn below.

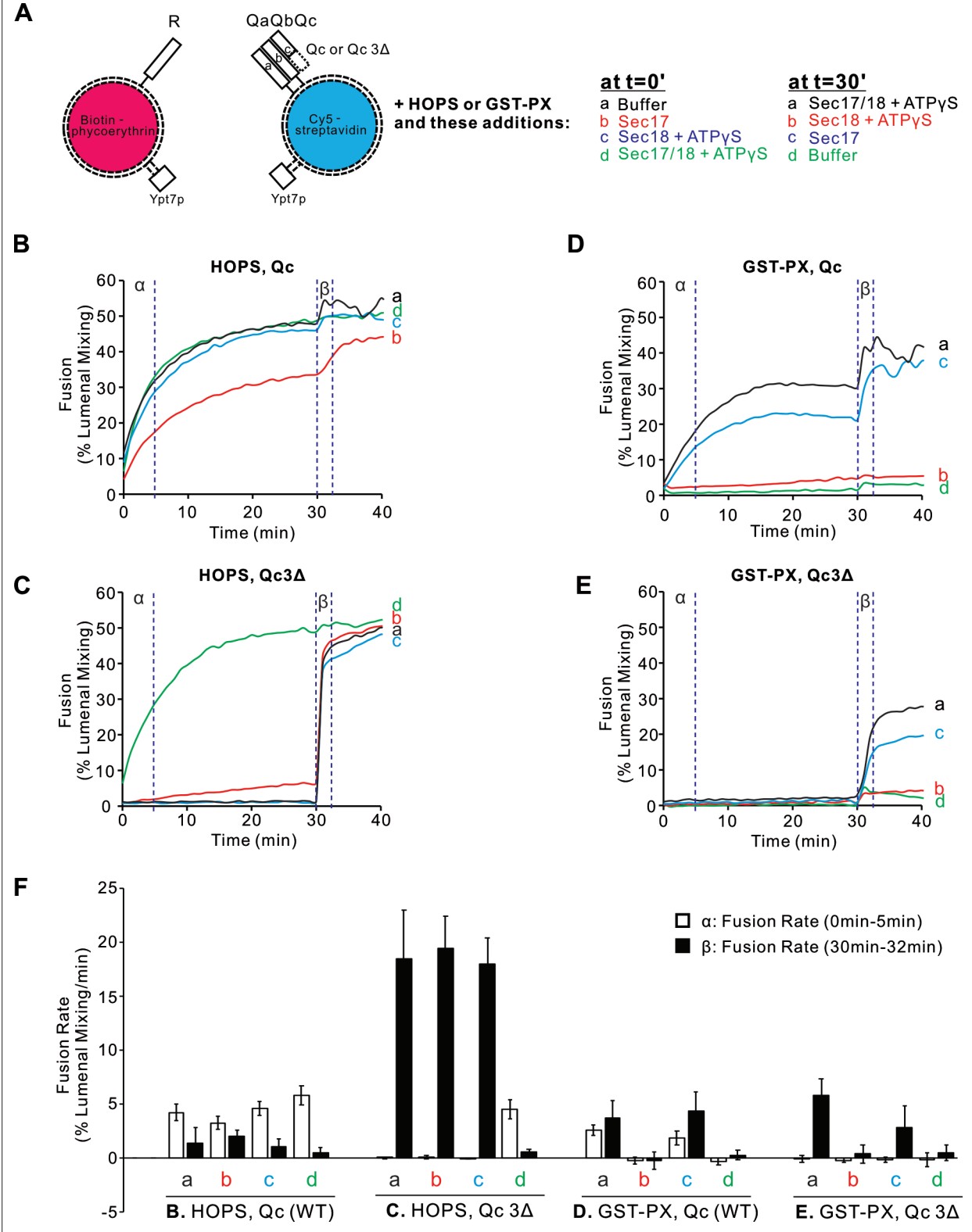

**Figure 1.** Sec17, Sec18, and ATP γ S promote rapid and homotypic fusion and vacuole protein sorting (HOPS)-independent fusion without SNARE zippering. Proteoliposomes were prepared with vacuolar lipids, membrane-anchored Ypt7, and either R or the 3Q SNAREs with molar ratios of 1Ypt7:8000 lipids and 1 of each SNARE/16,000 lipids. Since Qc3Δ is a labile member of SNARE complexes (**Song et al., 2021**), a fivefold molar excess to the other SNAREs was used in preparing Ypt7/QaQbQc3Δ proteoliposomes. The initial mixtures at t = 0 were 18 μl, and remaining components were added at 30 min in 2 μl. Vertical dotted lines delineate the first 5 min, termed α, and the 2 min β interval after further additions.

## HOPS is not required for Sec17/Sec18 stimulation of fusion

When the 3Q complex includes wild-type full-length Qc, HOPS-mediated fusion (*Figure 1B* α, black curve) shows only minor effects from adding either Sec17 (red), Sec18 with a nonhydrolyzable ATP analog (blue), or Sec17, Sec18, and ATPγS (green). In contrast, when the 3Q complex includes Qc3Δ to arrest SNARE zippering and block fusion (*Figure 1C* α, black curve), both Sec17 and Sec18 are required to bypass the zippering arrest and allow fusion (*Figure 1C* α, contrast the green curve vs. the blue, red or black curves).

The dimeric tether GST-PX (*Song and Wickner, 2019*) also supports fusion with pre-assembled wild-type Q SNAREs (*Figure 1D* α, black curve), but this fusion is blocked by Sec17 (*Figure 1D* α, red curve) without rescue by Sec18 (green curve). When fusion with the dimeric GST-PX tether is blocked by the Qc3Δ mutation (*Figure 1E* α, black curve), there is no rescue by Sec17, alone or in combination with Sec18/ATPγS (*Figure 1E* α, red and green curves), since HOPS is the only tether which bypasses inhibition by Sec17 (*Song and Wickner, 2019*).

At the end of these 30 min incubations, each reaction received a further addition of any components not added at time 0. After this addition, each incubation had Sec17, Sec18, and ATPγS. Fusion incubations continued in the β portion of the experiment, from 30 to 32 min, and beyond. Although full-length SNAREs support zippering and fusion with either the HOPS or GST-PX tether, a kinetic intermediate accumulates which gives some additional fusion upon addition of Sec17/Sec18/ATPγS (*Figure 1*, Bβ and Dβ, black curves). When SNARE zippering and the attendant fusion was blocked by the Qc3Δ mutation, HOPS-dependent zippering-bypass fusion requires Sec17, Sec18, and ATPγS (*Figure 1C* α), as reported (*Song et al., 2020*; *Song et al., 2021*). In their absence, fusion intermediate accumulated, since there is rapid fusion upon their addition (*Figure 1C* β, curves a–c). Strikingly, though the presence of Sec17 from the start of the incubation blocks the formation of rapid-fusion intermediate with the GST-PX tether (*Figure 1E* β, red curve), rapid-fusion intermediate does accumulate with the GST-PX tether when Sec17 is absent, as Sec17 addition at 30 min triggers fusion (*Figure 1E* β, curves a and c). Replicates of this experiment were quantified for the fusion rate during the α and β intervals (*Figure 1F*). Inhibition by Sec17 with the GST-PX tether instead of HOPS is only seen when Sec17 is present from the start of the incubation, prior to *trans*-SNARE assembly (*Figure 1E*, red curve b and green curve d), but once membranes have undergone tethering and *trans*-SNARE assembly, Sec17 and Sec18 support zippering-bypass fusion without HOPS (*Figure 1E* β, blue and black curves).

## Discussion

Sec17 may inhibit functional *trans*-SNARE complex assembly (*Wang et al., 2000*) but, once zippering has begun, Sec17 associates with partially zippered SNAREs to directly promote fusion (*Zick et al., 2015*; *Harner and Wickner, 2018*; *Song et al., 2021*). Sec17 can block fusion of reconstituted proteoliposomes without HOPS (*Mima et al., 2008*). HOPS engages each of the 4 SNAREs (*Stroupe et al., 2006*; *Baker et al., 2015*; *Song et al., 2020*) and catalyzes their assembly (*Baker et al., 2015*; *Orr et al., 2017*; *Jiao et al., 2018*; *Song et al., 2020*) in a manner which renders fusion resistant to Sec17 (*Mima et al., 2008*; *Xu et al., 2010*; *Song and Wickner, 2019*). Excessive Sec17 blocks the initial stages of fusion but stimulates fusion when added after *trans*-SNARE assembly (*Zick et al., 2015*). With the synthetic tether GST-PX, Sec17 which interacts with SNAREs before they assemble into *trans*-complexes inhibits the subsequent fusion (*Song and Wickner, 2019*). We now show that once fusion intermediates of *trans*-SNARE complexes have formed, Sec17 and Sec18 will promote zippering-bypass fusion whether tethering is by HOPS or by the synthetic tether GST-PX (*Figure 1C* β and Eβ). *Schwartz et al., 2017*, found that polyethylene glycol (PEG) and SNARE induced tethering and fusion, arrested by Qc3Δ, could be restored by added Sec17. PEG not only tethers, but may enhance SNARE-mediated fusion (*Zick and Wickner, 2013*) through promoting dehydration (*Lehtonen and Kinnunen, 1995*) and protein association (*Wälchli et al., 2020*) or, at high levels, induce fusion directly (*Burgess et al., 1992*). Dimeric GST-PX only acts as a tether. Our findings complement those of *Schwartz et al., 2017*, and show that a tether and partially zippered SNARE complex allow Sec17/Sec18 induction of fusion.

HOPS serves both as a tether (*Hickey and Wickner, 2010*) and as a catalyst to initiate SNARE assembly (*Baker et al., 2015*), but it has been unclear whether it catalyzes the later stages of zippering

or is needed for Sec17/Sec18-induced fusion without zippering. The engagement of SNAREs by HOPS (*Stroupe et al., 2006*; *Baker et al., 2015*; *Song et al., 2020*) largely bypasses Sec17 inhibition. Once SNAREs are partially zippered in trans, Sec17 and Sec18 do not need HOPS to support the completion of fusion. The completion of SNARE zippering is promoted by Sec17 (*Song et al., 2021*), and Sec17 displaces HOPS from SNARE complexes (*Collins et al., 2005*; *Schwartz et al., 2017*). The interactions among the SNAREs, Sec17/αSNAP, and Sec18/NSF are seen at a molecular level in the 20 s complex, consisting of a 4-SNARE coiled coil anchored to membranes at their C-termini, surrounded by up to 4 Sec17/αSNAP molecules, and all capped at the membrane-distal end by Sec18/NSF (*Zhao et al., 2015*). A 'trans-20s' (*Song et al., 2021*; *Rizo et al., 2021*) may generally drive fusion, since its elements are common to all exocytic and endocytic trafficking while the tethering complexes of other organelles are varied and often do not have the organelle's SM protein as a tightly bound subunit like Vps33 is in HOPS (*Baker and Hughson, 2016*). These findings support the model (*Schwartz et al., 2017*; *Song et al., 2021*) that HOPS only acts for tethering and to catalyze the initial phase of zippering in a Sec17-resistant manner. HOPS is specific for fusion at the vacuole/lysosome, but Sec17/αSNAP, Sec18/NSF, and SNAREs are general elements of exocytic and endocytic vesicular trafficking. Our current findings suggest that Sec17 and Sec18 support of zippering-bypass fusion may not be restricted to the vacuole/lysosome, but may contribute to SNARE-mediated fusion at other steps in endocytic and exocytic vesicular trafficking. Testing this hypothesis will necessitate the reconstitution of fusion at other organelles with their SNAREs, Rab, Rab-effector tethering complex, and SM protein.

# Materials and methods

## Key resources table

| Reagent type (species) or resource | Designation | Source or reference | Identifiers | Additional information |
|---|---|---|---|---|
| Gene (*Saccharomyces cerevisiae*) | Nyv1 | *Saccharomyces* Genome Database | SGD:S000004083 | |
| Gene (*Saccharomyces cerevisiae*) | Vam3 | *Saccharomyces* Genome Database | SGD:S000005632 | |
| Gene (*Saccharomyces cerevisiae*) | Vti1 | *Saccharomyces* Genome Database | SGD:S000004810 | |
| Gene (*Saccharomyces cerevisiae*) | Vam7 | *Saccharomyces* Genome Database | SGD:S000003180 | |
| Gene (*Saccharomyces cerevisiae*) | Ypt7 | *Saccharomyces* Genome Database | SGD:S000004460 | |
| Gene (*Saccharomyces cerevisiae*) | Sec17 | *Saccharomyces* Genome Database | SGD:S000000146 | |
| Gene (*Saccharomyces cerevisiae*) | Sec18 | *Saccharomyces* Genome Database | SGD:S000000284 | |
| Gene (*Saccharomyces cerevisiae*) | Vps33 | *Saccharomyces* Genome Database | SGD:S000004388 | |
| Gene (*Saccharomyces cerevisiae*) | Vps39 | *Saccharomyces* Genome Database | SGD:S000002235 | |
| Gene (*Saccharomyces cerevisiae*) | Vps41 | *Saccharomyces* Genome Database | SGD:S000002487 | |
| Gene (*Saccharomyces cerevisiae*) | Vps16 | *Saccharomyces* Genome Database | SGD:S000005966 | |
| Gene (*Saccharomyces cerevisiae*) | Vps11 | *Saccharomyces* Genome Database | SGD:S000004844 | |
| Gene (*Saccharomyces cerevisiae*) | Vps18 | *Saccharomyces* Genome Database | SGD:S000004138 | |
| Peptide, recombinant protein | R (Nyv1) | PMID:21976702 | | Purified from *Escherichia coli*. |
| Peptide, recombinant protein | Qa (Vam3) | PMID:18650938 | | Purified from *Escherichia coli*. |

*Continued on next page*

*Continued*

| Reagent type (species) or resource | Designation | Source or reference | Identifiers | Additional information |
|---|---|---|---|---|
| Peptide, recombinant protein | Qb (Vti1) | PMID:21976702 | | Purified from *Escherichia coli*. |
| Peptide, recombinant protein | Qc (Vam7) | PMID:17699614 | | Purified from *Escherichia coli*. |
| Peptide, recombinant protein | Qc (Vam7) 3Δ | PMID:19414611 | | Purified from *Escherichia coli*. |
| Peptide, recombinant protein | Ypt7-TM | PMID:31235584 | | Purified from *Escherichia coli*. |
| Peptide, recombinant protein | Sec17 | PMID:19414611 | | Purified from *Escherichia coli*. |
| Peptide, recombinant protein | Sec18 | PMID:8620540 | | Purified from *Escherichia coli*. |
| Peptide, recombinant protein | GST-PX | This Study | | Purified from *Escherichia coli*. |
| Peptide, recombinant protein | HOPS | PMID:18385512 | | Purified from *Saccharomyces cerevisiae* |
| Chemical compound,drug | Cy5-derivatized streptavidin | SeraCare Life Sciences | 5270–0023 | |
| Chemical compound,drug | Biotinylated PhycoE | Thermo Fisher Scientific | p811 | |
| Chemical compound,drug | Streptavidin | Thermo Fisher Scientific | 434302 | |
| Chemical compound,drug | 1,2-Dilinoleoyl-sn-glycero-3-phosphocholine | Avanti polar lipids | 850385 | |
| Chemical compound,drug | 1,2-Dilinoleoyl-sn-glycero-3-phospho-L-serine | Avanti polar lipids | 840040 | |
| Chemical compound,drug | 1,2-Dilinoleoyl-sn-glycero-3-phosphoethanolamine | Avanti polar lipids | 850755 | |
| Chemical compound,drug | 1,2-Dilinoleoyl-sn-glycero-3-phosphate | Avanti polar lipids | 840885 | |
| Chemical compound,drug | L-$\alpha$-Phosphatidylinositol | Avanti polar lipids | 840044 | |
| Chemical compound,drug | 1,2-Dipalmitoyl-*sn*-glycerol | Avanti polar lipids | 800816 | |
| Chemical compound,drug | Ergosterol | Sigma | 45480 | |
| Chemical compound,drug | PI(3)P | Echelon Bioscience | P-3016 | |
| Chemical compound,drug | Rhodamine DHPE | Invitrogen | L1392 | |
| Chemical compound,drug | NBD-PE | Invitrogen | N360 | |
| Chemical compound,drug | Marina-blue | Invitrogen | M12652 | |

Reagents were purchased, and proteins purified, as described in **Song et al., 2021**.

## GST-PX constructions

DNA encoding the PX domain from the Qc SNARE Vam7 (amino acyl residues 2–123) was amplified by PCR with CloneAMP HiFi PCR premix (Takara Bio USA, Mountain View, CA). The amplified DNA fragment was cloned into BamHI and SalI digested pGST parallel1 vector (**Sheffield et al., 1999**) with an NEBuilder HiFi DNA Assembly kit (NEB, Ipswich, MA).
    For GST-PX:

F AGGGCGCCATGGATCCGGCAGCTAATTCTGTAGGGAA.
R AGTTGAGCTCGTCGACTATGGCTTTGACAACTGCAGGA.
GST-PX was prepared as described (*Fratti and Wickner, 2007*).

Proteoliposome preparation and fusion assays were as described in *Song et al., 2021*. In brief, proteoliposomes were separately preincubated for 10 min at 27 °C with EDTA and GTP, followed by addition of $MgCl_2$, to load the Ypt7 with GTP. After separate preincubation for 10 min at 27 °C of both proteoliposome preparations and of mixtures of all soluble proteins (HOPS, GST-PX, Sec17, and Sec18/ATPγS) empty assay wells received in rapid succession 5 µl of Ypt7/R proteoliposomes, 5 µl of Ypt7/3Q proteoliposomes, and an 8 µl mixture of all soluble components. FRET representing fusion was recorded each minute for 30 min, as described (*Song et al., 2021*), then the multiwell plate was withdrawn and 2 µl of buffer, Sec17, Sec18, or a mixture of Sec17, Sec18, and Mg:ATPγS were added and the plate returned to the machine in time for the 31 min time-point and those thereafter. Concentrations in the 20 µl final reactions were 50 nM HOPS, 1.4 µM GST-PX, 500 nM Sec17, 250 nM Sec18, and 1 mM ATPγS where present.

## Acknowledgements

We thank Amy Orr for expert technical assistance and Alex Merz for fruitful discussions. This work was supported by NIGMS grant R35GM118037.

## Additional information

### Funding

| Funder | Grant reference number | Author |
| --- | --- | --- |
| National Institute of General Medical Sciences | R35GM118037 | William T Wickner |

The funders had no role in study design, data collection and interpretation, or the decision to submit the work for publication.

### Author contributions

Hongki Song, Conceptualization, Data curation, Visualization, Writing - original draft, Writing - review and editing; William T Wickner, Conceptualization, Funding acquisition, Resources, Supervision, Validation, Writing - original draft, Writing - review and editing

### Author ORCIDs

Hongki Song ⓘ http://orcid.org/0000-0002-3761-5434
William T Wickner ⓘ http://orcid.org/0000-0001-8431-0468

### Decision letter and Author response

Decision letter https://doi.org/10.7554/eLife.73240.sa1
Author response https://doi.org/10.7554/eLife.73240.sa2

## Additional files

### Supplementary files

• Transparent reporting form

### Data availability

Figure 1 - Source Data 1 contains the numerical data used to generate figure 1.

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
