## [Editor Report]

This Research Advance answers an important question arising from the author's previous paper on in vitro vacuole fusion: whether the tether HOPS is needed for an unexpected direct role played by Sec17 and Sec18 in facilitating SNARE zippering and thus membrane fusion. An artificial tether substitutes for HOPS, so HOPS is not specifically needed. Therefore, this unexpected role for Sec17/18 could extend to other SNARE-mediated fusion reactions.

---

## [Decision Letter]

[Editors’ note: the authors submitted for reconsideration following the decision after peer review. What follows is the decision letter after the first round of review.]

Thank you for submitting the paper "Membrane Fusion can be Driven by Sec18/NSF, Sec17/αSNAP, and *trans*-SNARE complex without HOPS" for consideration at *eLife*. Your initial submission has been assessed by a Senior Editor in consultation with expert members of the Board of Reviewing Editors. Although the work is of interest, we regret to inform you that the findings at this stage are too preliminary for further consideration at *eLife*.

Expert #1. "…As a "research advance" the work does answer an important question left by their previous *eLife* paper: whether HOPS is needed for Sec17 and Sec18 to engage partially zippered SNAREs. An artificial tether substitutes so HOPS is not specifically needed and this unexpected role for Sec17/18 could extend to other fusion reactions. It is careful biochemistry that extends an e*Life* paper and provides data supporting a (controversial) direct role for Sec17/18 in content mixing. However, in its present form, the work doesn't go far enough to address whether the proposed novel role for Sec17/18 occurs and makes an important contribution to the reaction under normal conditions."

Expert #2. "Here the authors rely on an artificial truncation of one of the SNAREs, as in the previous paper, and eliminate HOPS in favor of an artificial tether. This creates a situation in which Sec17 and 18 appear to be essential for fusion. How specific is the requirement for Sec17 and 18 when their natural binding partners are altered? For this study to be meaningful, the authors would have to show that the pair of native proteins are required for fusion per se under more normal conditions with the full set of SNAREs, HOPS etc."

[Editors’ note: further revisions were suggested prior to acceptance, as described below.]

Thank you for submitting your article "Fusion of Tethered Membranes can be Driven by Sec18/NSF and Sec17/αSNAP without HOPS" for consideration by *eLife*. Your article has been reviewed by 3 peer reviewers, one of whom is a member of our Board of Reviewing Editors, and the evaluation has been overseen by Suzanne Pfeffer as the Senior Editor. The following individual involved in review of your submission has agreed to reveal their identity: Alexey J Merz (Reviewer #3).

Essential revisions:

This brief study has now been reviewed by three experts. There was only one technical issue raised (titration of components). However, all three reviewers were concerned about the "incremental" nature of the advance. The study shows that their Sec17/18-dependent vacuole fusion reaction employing arrested SNAREs and without ATP hydrolysis proceeds without a specific contribution from HOPS. The potential significance of this to a wider audience is that Sec17/18 may contribute directly to the fusion step of many or all SNARE-mediated fusion reactions. But this aspect is not sufficiently tested and needs to be addressed. This and the other reviewer issues are compiled below. Feasibility is clearly a concern regarding an in vivo test (point#4) but the issue of evidence that it normally occurs (e.g. rate acceleration for wildtype components as in point#1) merits particular attention.

1. A main issue is the increment of advance. It succeeds in filling in a missing piece of their 2021 paper but on its own does not substantially advance understanding of the critical issue of whether Sec17/18 normally plays this role. If it did, one would expect fusion by normal SNAREs to be significantly accelerated by Sec17/18. From Figure 1B, 1F, this does not appear to be the case.

2. One would expect Sec17/18 association with the normal SNAREs to occur prior to fusion, but there is no data to address this.

3. The concentration of Sec17/18, the most critical component, is not indicated in either the figure legend or methods.

4. Can this unexpected role for Sec17/18 be demonstrated/confirmed for intact cells? Is it essential for wildtype rates of fusion? Or is it strictly redundant in the face of the assembly force of SNARE pairing as a driving force for fusion?

5. Does it actually act on other SNAREs? This is important because the clearest argument for this brief study being viewed as a highly significant research advance (over their previous work) is the implication that the unexpected direct role of Sec17/18 can now be considered likely to take place in other, if not all, SNARE-mediated fusion events.

6. No concentrations for any soluble component are given. The final concentrations of Vam7/Qc, Sec17, Sec18, HOPS, ATPgS, etc., must be specified. Readers should not have to look at prior publications to locate this information. A possible criticism of this and some similar prior work is that the experiments are conducted at only single concentrations of each reagent, rather than using concentration titrations for the protein components in the reaction. Nevertheless, the results support the authors' conclusions and are broadly consistent with prior work.

7. The last paragraph of the introduction could probably be omitted without loss of clarity.

8. Line 46: extensive mutational analyses (both in vitro and in vivo) of the Sec17 membrane-interaction loop is present in Schwartz et al. 2017. This work should be cited.

9. Lines 62-65: the pioneering paper by Hanson, Jahn et al. was a key step toward the consensus that SNARE complexes are oriented in a parallel coiled coil, and that Sec17/18 may act after, rather than before, fusion. It should be cited.

10. Lines 72-73: again, Schwartz et al., 2017 should be cited.

---

## [Author Response]

[Editors’ note: the authors resubmitted a revised version of the paper for consideration. What follows is the authors’ response to the first round of review.]

Expert #1. "…As a "research advance" the work does answer an important question left by their previous eLife paper: whether HOPS is needed for Sec17 and Sec18 to engage partially zippered SNAREs. An artificial tether substitutes so HOPS is not specifically needed and this unexpected role for Sec17/18 could extend to other fusion reactions. It is careful biochemistry that extends an eLife paper and provides data supporting a (controversial) direct role for Sec17/18 in content mixing. However, in its present form, the work doesn't go far enough to address whether the proposed novel role for Sec17/18 occurs and makes an important contribution to the reaction under normal conditions."Expert #2. "Here the authors rely on an artificial truncation of one of the SNAREs, as in the previous paper, and eliminate HOPS in favor of an artificial tether. This creates a situation in which Sec17 and 18 appear to be essential for fusion. How specific is the requirement for Sec17 and 18 when their natural binding partners are altered? For this study to be meaningful, the authors would have to show that the pair of native proteins are required for fusion per se under more normal conditions with the full set of SNAREs, HOPS etc."

We now state clearly, with reference, that Sec17/Sec18 stimulation has been reported with all wild-type components, i.e., SNAREs. Our new finding establishes that the role of Sec17/Sec18 in promoting fusion is not limited to HOPS-catalyzed reactions, and thus merits serious considerations for other endocytic and exocytic trafficking steps

[Editors’ note: what follows is the authors’ response to the second round of review.]

Essential revisions:This brief study has now been reviewed by three experts. There was only one technical issue raised (titration of components). However, all three reviewers were concerned about the "incremental" nature of the advance. The study shows that their Sec17/18-dependent vacuole fusion reaction employing arrested SNAREs and without ATP hydrolysis proceeds without a specific contribution from HOPS. The potential significance of this to a wider audience is that Sec17/18 may contribute directly to the fusion step of many or all SNARE-mediated fusion reactions. But this aspect is not sufficiently tested and needs to be addressed. This and the other reviewer issues are compiled below. Feasibility is clearly a concern regarding an in vivo test (point#4) but the issue of evidence that it normally occurs (e.g. rate acceleration for wildtype components as in point#1) merits particular attention.1. A main issue is the increment of advance. It succeeds in filling in a missing piece of their 2021 paper but on its own does not substantially advance understanding of the critical issue of whether Sec17/18 normally plays this role. If it did, one would expect fusion by normal SNAREs to be significantly accelerated by Sec17/18. From Figure 1B, 1F, this does not appear to be the case.

This point is now directly addressed in the next-to-last paragraph of the Introduction:

"Sec17 and Sec18 enhance the rate of HOPS-dependent vacuole membrane fusion with all wild-type fusion components (Mima, 2008, Zick et al., 2015; Song et al., 2017; Schwartz et al., 2017) without requiring ATP hydrolysis (Zick et al., 2015; Song et al., 2017). […] Our current work, substituting a synthetic protein tether for HOPS, shows that Sec17/Sec18 can also drive HOPS-independent fusion."

A similar analysis in non-vacuolar fusion reactions will require their reconstitution with all purified components, not only SNAREs, NSF, and SNAP but also SM protiens, tethering complexes, Rabs, and other factors. This is nearly achieved now for synaptic fusion, but the multiple proteins special to the Ca-responsivity of the synapse (synaptotagmin, Munc13, Complexin) distinguish its fusion from that of general endocytic and exocytic traffic. Endosome homotypic fusion was brilliantly reconstituted by Zerial and colleagues in a Nature study, but there has been no follow-through of mechanistic studies in the ensuing years. Thus other systems to test the generality of our findings with vacuolar fusion are still lacking.

2. One would expect Sec17/18 association with the normal SNAREs to occur prior to fusion, but there is no data to address this.

This was addressed most directly in Harner and Wickner (2018), and we've added this at the start of the Discussion in the revised paper.

3. The concentration of Sec17/18, the most critical component, is not indicated in either the figure legend or methods.

Sorry for this awful omission. We've now added these concentrations to the figure legend: 500nM Sec17, 250nM Sec18, and 1mM ATPgS.

4. Can this unexpected role for Sec17/18 be demonstrated/confirmed for intact cells? Is it essential for wildtype rates of fusion? Or is it strictly redundant in the face of the assembly force of SNARE pairing as a driving force for fusion?

We now cite this more clearly in the introduction: "Rapid HOPS-dependent fusion requires both SNARE zippering and Sec17 with its apolar loops, as analyzed in vitro and in vivo, where overexpression of Sec17 with an intact N-terminal apolar loop substantially restored the fusion defect caused by Qc-SNARE domain truncation (Schwartz et al., 2017)." Is it essential for wildtype rates of fusion? Or is it strictly redundant in the face of the assembly force of SNARE pairing as a driving force for fusion? I'm not sure how we'd measure the wild-type (in vivo) rate of fusion, but we certainly haven't in the past. I'm a bit unclear about just what's being asked here. Zippering and Sec17-induced fusion are redundant, but (perhaps more important) they're complementary. The Song et al. 2021 *eLife* paper, which the current submission is built upon, shows that " Slow fusion can be driven either by zippering or by the apolar loops of Sec17 which had assembled on a platform of at least partially zippered *trans*-SNARE complex, but rapid HOPS-dependent fusion requires both SNARE zippering and Sec17 with its apolar loops." (as stated in the current Introduction).

5. Does it actually act on other SNAREs? This is important because the clearest argument for this brief study being viewed as a highly significant research advance (over their previous work) is the implication that the unexpected direct role of Sec17/18 can now be considered likely to take place in other, if not all, SNARE-mediated fusion events.

We now address this important question directly at the end of the Discussion: "Our current findings suggest that Sec17 and Sec18 support of zippering-bypass fusion may not be restricted to the vacuole/lysosome, but may contribute to SNARE-mediated fusion at other steps in endocytic and exocytic vesicular trafficking. Testing this hypothesis will necessitate the reconstitution of fusion at other organelles with their SNAREs, Rab, Rab-effector tethering complex, and SM protein." This is far beyond our current scope.

6. No concentrations for any soluble component are given. The final concentrations of Vam7/Qc, Sec17, Sec18, HOPS, ATPgS, etc., must be specified. Readers should not have to look at prior publications to locate this information. A possible criticism of this and some similar prior work is that the experiments are conducted at only single concentrations of each reagent, rather than using concentration titrations for the protein components in the reaction. Nevertheless, the results support the authors' conclusions and are broadly consistent with prior work.

I apologize for this awful omission. These concentrations are now explicit in the last paragraph of the Methods. All the assays are with 3Q-proteoliposome, and thus there is no concentration given for Qc (which isn't added per se, it's part of the 3Q).

7. The last paragraph of the introduction could probably be omitted without loss of clarity.

We have rearranged, and rewritten, the Introduction with a goal of added clarity. We beg to be allowed to keep the last paragraph, as we feel it provides a useful overview of the strategy we've followed.

8. Line 46: extensive mutational analyses (both in vitro and in vivo) of the Sec17 membrane-interaction loop is present in Schwartz et al. 2017. This work should be cited.

Thank you, we do now.

9. Lines 62-65: the pioneering paper by Hanson, Jahn et al. was a key step toward the consensus that SNARE complexes are oriented in a parallel coiled coil, and that Sec17/18 may act after, rather than before, fusion. It should be cited.

Thank you, we've added two citations to their pioneering work.

10. Lines 72-73: again, Schwartz et al., 2017 should be cited.

Thank you, now done.